



# 500-thousand-year-old basal ice at Skytrain Ice Rise, West Antarctica, estimated with the $^{36}$Cl/$^{10}$Be ratio

Niklas Kappelt[1], Eric Wolff[2], Marcus Christl[3], Christof Vockenhuber[3], Philip Gautschi[3], and Raimund Muscheler[1]

[1]Department of Geology, Lund University, Sölvegatan 12, 22362 Lund
[2]Department of Earth Sciences, University of Cambridge, Downing Street, CB2 3EQ Cambridge, United Kingdom
[3]Laboratory of Ion Beam Physics, Swiss Federal Institute of Technology, Otto-Stern Weg 5, 8093 Zürich, Switzerland

**Correspondence:** Niklas Kappelt (niklas.kappelt@geol.lu.se)

**Abstract.** Dating the bottommost section of an ice core is often complicated by strong layer thinning and possible disturbances in the stratigraphy. The radioactive decay of atmospherically produced $^{36}$Cl and $^{10}$Be can provide age estimates, where traditional methods can no longer be used. In this study, we investigated ice from the bottom of the Skytrain ice core, which was drilled in West Antarctica next to the Ronne Ice Shelf and has previously been dated to 126 kyr BP about 24 m above bedrock.

Apart from decay, radionuclide concentrations in ice can be influenced by production rate variations, atmospheric transport and deposition variations, and, at low accumulations sites, by chlorine loss through hydrogen chloride outgassing. Using the $^{36}$Cl/$^{10}$Be ratio largely removes production related variations and we were able to confirm that no $^{36}$Cl loss occurs at Skytrain Ice Rise, as the nuclear weapon test caused peak in $^{36}$Cl concentrations was found at the expected depth corresponding to the 1950s and 60s. An analysis of samples with known age showed that individual radionuclide concentrations and the $^{36}$Cl/$^{10}$Be ratio are negatively correlated to the $\delta^{18}$O signal, which was used to apply a climate correction that enabled a higher precision for age estimates of previously undated samples. The deepest analysed section of the Skytrain ice core was found to be $552 \pm 112$ kyr old.

## 1 Introduction

In 2018/2019, an ice core was drilled at Skytrain Ice Rise in West Antarctica, adjacent to the Ronne Ice Shelf (Mulvaney et al., 2021). A chronology extending to 126 kyr BP at a depth of 627 m was established and the core was used to constrain the retreat of the West Antarctic Ice Sheet in the early Holocene and analyse its stability in the last interglacial period (Hoffmann et al., 2022; Mulvaney et al., 2023; Grieman et al., 2024). With a total length of 651 m, however, 24 m of ice at the bottom remain undated. Around the last interglacial, the record was found to be discontinuous with possible flow disturbances and folding, making the ice especially challenging to date with traditional dating methods (Mulvaney et al., 2023). Radionuclides have the
potential to provide independent age estimates, making it possible to verify and extend the existing chronology. The $^{36}$Cl/$^{10}$Be





ratio has an effective half-life of 384 kyr (Audi et al., 2017; Chmeleff et al., 2010) and is largely independent of the cosmic ray flux, which governs the production rates of individual cosmogenic radionuclides (Wagner et al., 2000; Kappelt et al., 2025). However, there are three processes that can cause its values in ice to deviate from the expected production rate ratio of 0.0875

(Poluianov et al., 2016).

First, $^{36}$Cl can be lost from the firn due to the mobility of hydrogen chloride (HCl) gas, which should only be a concern at low accumulation sites, such as Vostok, EPICA Dome C and Little Dome C (Delmas et al., 2004; Kappelt et al., 2025). Nuclear bomb tests in the 1950s and 60s led to a peak in the $^{36}$Cl concentration, which was measured in several ice cores from Greenland and Antarctica and returned to natural levels by the 1980s (Synal et al., 1990; Heikkilä et al., 2009). At Vostok (accumulation

rate 2.1 $\mathrm{g\,cm^{-2}\,yr^{-1}}$ (Ekaykin et al., 2002)), the peak is shifted upwards with elevated $^{36}$Cl concentrations remaining for decades (Delmas et al., 2004; Pivot et al., 2019). A minimum threshold of 4 - 8 $\mathrm{g\,cm^{-2}\,yr^{-1}}$ ice accumulation rate has been suggested to be required for the preservation of sea-salt chlorine, which can gas out as HCl as well (Röthlisberger et al., 2003; Benassai et al., 2005). At Skytrain ice rise, the near-surface accumulation rate is 13.5 $\mathrm{g\,cm^{-2}\,yr^{-1}}$ (Hoffmann et al., 2022), so no loss is expected.

The second challenge for radionuclide dating is the climate related variability of the $^{36}$Cl/$^{10}$Be ratio recorded in ice cores. For example, in the EPICA Dome C (EDC) ice core, a one-$\sigma$ standard deviation of 33 % from the mean was observed for decay corrected glacial samples, which are presumably free of $^{36}$Cl loss (Kappelt et al., 2025), hinting towards processes other than decay influencing the recorded signal. Generally, the measured ratio in polar ice cores is higher than the calculated atmospheric production rate ratio of 0.0875 (Poluianov et al., 2016): the average decay-corrected $^{36}$Cl/$^{10}$Be ratio ranges from 0.092 to 0.12 in

Antarctic ice cores and from 0.12 to 0.22 in Greenland ice cores, excluding samples affected by $^{36}$Cl loss (Kappelt et al., 2025; Kanzawa et al., 2021; Yiou et al., 1997; Baumgartner et al., 1997a; Muscheler et al., 2004; Wagner et al., 2001; Muscheler, 2000; Lukasczyk, 1994; Elmore et al., 1987). Both, the variability and the mean ratio are likely a product of different transport and deposition pathways for the two radionuclides, which are primarily produced in the stratosphere. However, $^{10}$Be exclusively attaches to aerosols, while $^{36}$Cl can attach to aerosols or form H$^{36}$Cl gas, which likely is its predominant state (Zerle et al.,

1997; Heikkilä et al., 2009; Zheng et al., 2025). To improve age estimates, the variability must be explained, which requires a better understanding of the transport and deposition processes and how they are affected by climatic changes. In the GRIP (Greenland ice core project) ice core, higher $^{36}$Cl/$^{10}$Be ratios can be found in the last glacial period than in the Holocene (Yiou et al., 1997). Discrete samples from the EDC ice core also hint towards a correlation with the deuterium isotope signal, which is indicative of the temperature and accumulation rate ($r = 0.70$, if two samples with exceptionally high $^{36}$Cl/$^{10}$Be ratios from

the Eemian are excluded from the dataset (Kappelt et al., 2025)). However, interglacial $^{36}$Cl loss may drive the correlation at EDC and the GRIP data requires a more in-depth analysis to confirm a potential correlation with water isotopes.

The third process that can affect the $^{36}$Cl/$^{10}$Be ratio is post-depositional $^{10}$Be mobility. The radionuclide has been shown to increasingly associate with dust in the deepest section of the GRIP ice core (Baumgartner et al., 1997b), $^{10}$Be concentrations spikes and an apparent depletion were found in the EDC core (Raisbeck et al., 2006; Kappelt et al., 2025), and a large variability

in the deepest part of the EPICA Dronning Maud Land (EDML) core has been suggested to be caused by $^{10}$Be mobilisation (Auer et al., 2009). While the environmental conditions at the three drill sites are rather different, the affected core sections



are all from depths below 2700 m. In comparison, the Skytrain ice core is comparably short with a length of 651 m, so $^{10}$Be mobility may not be an issue, although it is not clear yet whether depth or age is the more important factor.

In this study, we estimated the age of five samples from the undated bottom section of the Skytrain ice core with the $^{36}$Cl/$^{10}$Be ratio. We found a monotonically decreasing ratio with depth suggesting about half a million year old ice at the very bottom. To confirm that no $^{36}$Cl loss occurs at Skytrain ice rise, we first measured radionuclide concentrations from recent decades and compared the $^{36}$Cl peak from nuclear weapon tests with data from other sites. Then, samples from the Holocene and the last glacial period were analysed to find possible correlations between the $^{36}$Cl/$^{10}$Be ratio and different climate proxies, and establish a mean value serving as a starting point to determine the degree of radioactive decay in undated samples.

## 2   Methods

### 2.1   Sample selection

Two sets of radionuclide samples from the Skytrain ice core were analysed. We obtained 31 radionuclide samples with annual resolution for the years 1982 to 2013 and 28 samples with biennial resolution for the years 1954 to 1982 with the aim to assess whether the $^{36}$Cl bomb peak is found at the expected depth. The respective layers were located based on the ST22

chronology, which was developed using annual layer counting in this depth section and contains uncertainties of 1.0 to 2.3 yr for the identified years in this age range (Hoffmann et al., 2022). The second dataset was obtained with the aim of analysing the climate-related long-term variability of the $^{36}$Cl/$^{10}$Be ratio and to estimate the age at the bottom of the ice core. It contains six discrete samples from the Holocene, twelve samples from the last glacial period, and five samples from depths below the dated section.

### 2.2   Radionuclide measurements

All radionuclide samples were prepared following the procedures described by Adolphi et al. (2014) and Delmas et al. (2004) with the addition of 0.299 mg $^{9}$Be carrier (Adolphi et al., 2014; Delmas et al., 2004). To the ice samples from recent decades, 2.00 mg of stable chlorine carrier ($^{35}$Cl and $^{37}$Cl) was added, while 4.00 mg was added to deeper samples.

    The $^{10}$Be and $^{36}$Cl concentrations were determined with Accelerator Mass Spectrometry (AMS) at the Laboratory for Ion

Beam Physics at ETH Zurich. $^{10}$Be/$^{9}$Be ratios were normalised to the ETH in-house standard S2007N with a nominal ratio of $^{10}$Be/$^{9}$Be $= (28.1 \pm 0.8) \cdot 10^{-12}$, which in turn is normalised against the ICN 01-5-1 standard with a nominal value of $^{10}$Be/$^{9}$Be $= 2.709 \cdot 10^{-11}$ (Nishiizumi et al., 2007; Christl et al., 2013). All $^{36}$Cl samples were normalised with the in-house standard K382/4N (Christl et al., 2013), which has a nominal value of $^{36}$Cl/Cl $= (17.36 \pm 0.34) \cdot 10^{-12}$. All $^{10}$Be and $^{36}$Cl samples were blank corrected with Milli-Q water blanks prepared in parallel to the respective samples.

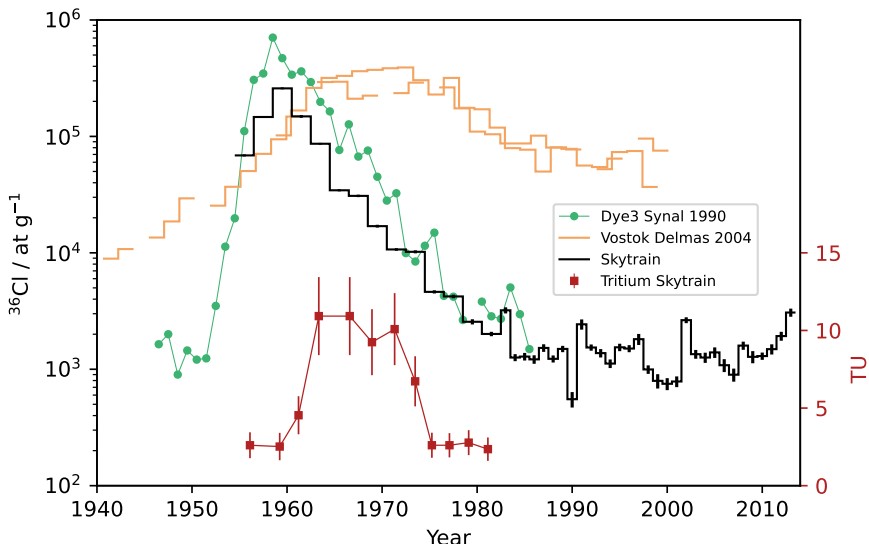

**Figure 1.** Comparison of the Skytrain $^{36}$Cl concentration (black) and tritium data (red) with $^{36}$Cl data from Dye3 in Greenland (green) and Vostok in East Antarctica (orange) over the last decades (Synal et al., 1990; Delmas et al., 2004; Baroni et al., 2011; Hoffmann et al., 2022).

## 3    Results and Discussion

### 3.1    Bomb peak data

Figure 1 depicts $^{36}$Cl concentrations and tritium data from recent decades in the Skytrain ice core together with $^{36}$Cl data from an ice core drilled close to the Dye3 site in Greenland (Synal et al., 1990) and a snow pit near Vostok in East Antarctica (Delmas et al., 2004). While the Dye3 data was published with an age scale based on annual layer counting with $H_2O_2$ (Synal et al., 1990), we adapted a timescale from Baroni et al. (2011) for the Vostok data. It is based on two types of absolute time markers: non-sea-salt sulphate layers corresponding to the volcanic Agung (1963) and Pinatubo (1991) eruptions as well as beta activity peaks from nuclear weapon tests (1955 and 1965), which are related to short-lived radionuclides and independent of $^{36}$Cl (Baroni et al., 2011; Pourchet et al., 2003). The timescale was originally developed for another snow pit nearby, but has been used for a similar comparison before (Pivot et al., 2019) and is sufficient for our qualitative comparison between low and high accumulation sites.

The $^{36}$Cl concentration at Skytrain strongly resembles that of Dye3, peaking in the late 1950s and returning to a natural magnitude by the mid-1980s. This is the expected fallout from the $^{36}$Cl produced by nuclear weapon tests and in agreement with modelled deposition fluxes as well as observations in other polar and alpine ice cores (Heikkilä et al., 2009). In contrast, the $^{36}$Cl concentration at Vostok peaks several years later, remains high for a decade and does not return to natural levels even in the late 1990s (Delmas et al., 2004). This has been attributed to a release of H$^{36}$Cl gas during snow metamorphism, which




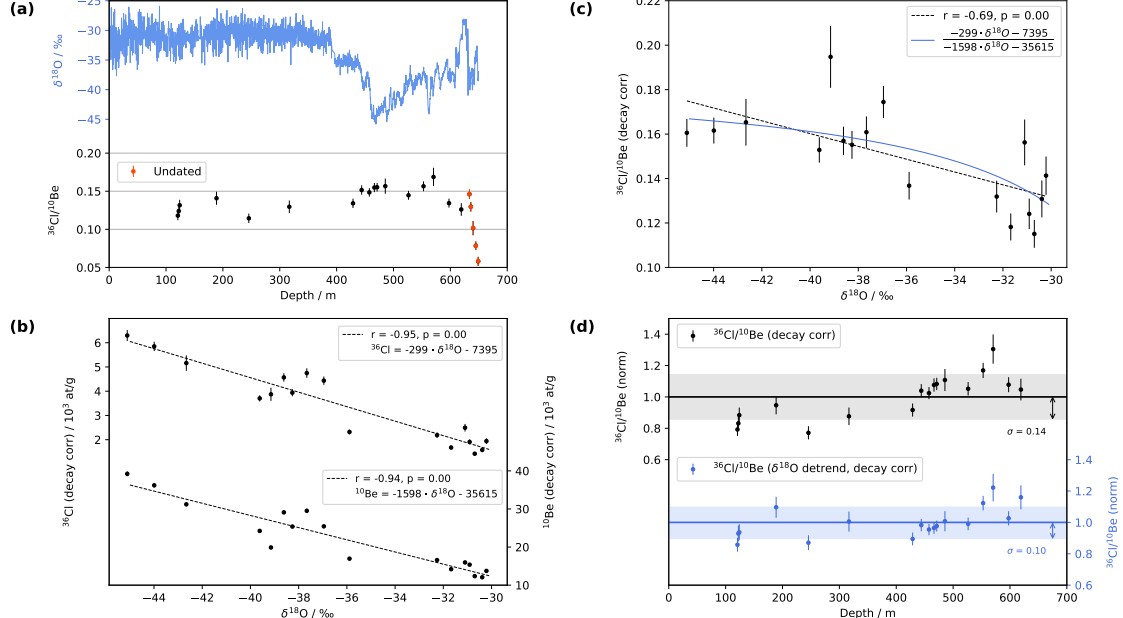

**Figure 2.** (A) $\delta^{18}$O isotope data from Mulvaney et al. (2023) and measured $^{36}$Cl/$^{10}$Be ratio (Mulvaney et al., 2023). (B) Relationship between the decay corrected individual radionuclide concentrations and $\delta^{18}$O. (C) Relationship between the decay corrected $^{36}$Cl/$^{10}$Be ratio and $\delta^{18}$O. (D) Comparison of the normalised, decay corrected $^{36}$Cl/$^{10}$Be ratio with the decay corrected $^{36}$Cl/$^{10}$Be ratio, detrended with the non-linear $\delta^{18}$O fit of panel C. The latter has a lower standard deviation from the mean.

can lead to re-deposition in upper layers or loss of $^{36}$Cl. The comparison shows that $^{36}$Cl concentrations in the Skytrain ice core are not affected by this mobility, providing suitable conditions for age estimates with the $^{36}$Cl/$^{10}$Be ratio and an analysis of climate-related variability.

Tritium ($^{3}$H) is another product of nuclear weapon tests and was used to constrain the Skytrain age scale by visually matching
the peak in the ice core with peak concentrations in precipitation samples from New Zealand (Hoffmann et al., 2022). Elevated Tritium is found at shallower depths than $^{36}$Cl, which was produced by neutron activation of $^{35}$Cl from seawater sodium chloride (Delmas et al., 2004). The largest quantities were therefore produced during tests on ships and small islands in 1954, 1956, and 1958 (Synal et al., 1990; Heikkilä et al., 2009). Most tritium, on the other hand, was released in the early 60s, when the tests with the highest total yield were carried out in the northern hemisphere, leading to broad peak in the southern
hemisphere (UNSCEAR 2000, p. 207; Fourré et al. 2006)

## 3.2 Climate influence

The $^{36}$Cl/$^{10}$Be ratio measured in ice from the Holocene and the last glacial period as well as ice from below the dated section of the ST22 chronology is shown in Figure 2A together with the $\delta^{18}$O signal from (Mulvaney et al., 2023). As shown in panel





C, there is a negative correlation between the decay corrected $^{36}$Cl/$^{10}$Be ratio and the $\delta^{18}$O signal with a Pearson's correlation

coefficient of $r = -0.69$ and $p < 0.005$. Note that the applied decay correction is relatively small ($< 25\,\%$), as the dated section is relatively young compared to the 384 kyr half-life. The individual radionuclide concentrations correlate more strongly with the $\delta^{18}$O signal (Figure 2B, $r = -0.95$ for $^{36}$Cl and $r = -0.94$ for $^{10}$Be), which may be caused by dilution: higher accumulation rates lead to lower radionuclide concentrations. The accumulation rate drastically changes between glacial and interglacial periods and can be estimated proportional to $\exp(\delta^{18}$O$)$ (Parrenin et al., 2007), so the correlation should be tested between the

natural logarithm of the concentrations and the $\delta^{18}$O signal, which barely changes the correlation coefficients ($r = -0.95$ for $^{36}$Cl and $r = -0.94$ for $^{10}$Be). The requirement for a dilution effect is a radionuclide deposition flux which is independent or only weakly dependent of the the accumulation rate (Baumgartner et al., 1997a). This is the case at EDC and Dome Fuji, where $^{10}$Be is deposited predominantly dry, i.e. not through precipitation (Raisbeck and Yiou, 1985; Cauquoin et al., 2015; Sasa et al., 2010; Horiuchi et al., 2008). However, a dilution effect was also observed at the Das2 and GRIP sites in Greenland, where

radionuclide deposition occurs mainly through precipitation (wet) (Baumgartner et al., 1997a; Pedro et al., 2012). The effect can be explained by a progressive temporary depletion of local atmospheric radionuclide concentrations during a precipitation event, leading to a dilution from continued precipitation (Ishikawa et al., 1995). Dilution can therefore still occur at Skytrain Ice Rise, where radionuclides are expected to be deposited predominantly wet (Heikkilä and Smith, 2013). In this case, the effect on $^{36}$Cl and $^{10}$Be concentrations would be identical, but different slopes are obtained for the correlations of normalised

radionuclide concentrations with $\delta^{18}$O, (-0.0874 ± 0.0077) $‰^{-1}$ for $^{36}$Cl and (-0.0720 ± 0.0069) $‰^{-1}$ for $^{10}$Be. Therefore, the relationship with the $\delta^{18}$O signal is likely caused in part by dilution with an additional contribution from the mechanisms proposed below.

        Increased precipitation in warmer periods may also lead to increased rainout (in-cloud) and washout (below-cloud) effects, depleting atmospheric radionuclide concentrations en-route from mid-latitudes, where radionuclide-rich, stratospheric air en-

ters the troposphere, towards the poles (Pedro et al., 2012, 2011; Muscheler, 2000). Since we observe lower ratios in warmer periods, the depletion would have to be stronger for $^{36}$Cl than for $^{10}$Be. Measurements of the $^{36}$Cl/$^{10}$Be ratio in precipitation samples, on the other hand, showed an increasing ratio throughout the course of a given precipitation event, suggesting a faster removal of $^{10}$Be instead (Knies et al., 1994; Lukasczyk, 1994). However, the only two studies of this kind were conducted in Indiana in the United States and in Switzerland, so their results may not be applicable for precipitation over the Southern

Ocean.

        Another hypothesis for the $^{18}$O correlation relates to a recent modelling study, in which Zheng et al. (2025) explain $^{36}$Cl/$^{10}$Be ratio values above the production rate ratio in polar deposition fluxes with a higher scavenging efficiency for $^{36}$Cl than $^{10}$Be in mixed-phase (water and ice) clouds (Zheng et al., 2025; Stier et al., 2005; Stuart and Jacobson, 2003). A higher ice content in colder periods would amplify the effect, in agreement with higher $^{36}$Cl/$^{10}$Be ratios in colder periods observed in the Skytrain,

GRIP and EDC ice cores (Kappelt et al., 2025; Yiou et al., 1997; Wagner et al., 2001; Muscheler et al., 2004). The correlation with the $\delta^{18}$O signal would then relate to temperature as well as the accumulation rate.

        The strong correlations between the individual radionuclide concentrations and $\delta^{18}$O imply that the relationship between the $^{36}$Cl/$^{10}$Be ratio and $\delta^{18}$O may not be linear, but rather the product of individual correlations. It can be expressed as





$$\frac{^{36}\mathrm{Cl}}{^{10}\mathrm{Be}}(\delta^{18}\mathrm{O}) = \frac{\frac{\Delta^{36}\mathrm{Cl}}{\Delta\delta^{18}\mathrm{O}} \cdot \delta^{18}\mathrm{O} + b_{^{36}\mathrm{Cl}}}{\frac{\Delta^{10}\mathrm{Be}}{\Delta\delta^{18}\mathrm{O}} \cdot \delta^{18}\mathrm{O} + b_{^{10}\mathrm{Be}}} = \frac{-299 \cdot \delta^{18}\mathrm{O} - 7390}{-1598 \cdot \delta^{18}\mathrm{O} - 35600}, \tag{1}$$

resulting in a decreasing $^{36}\mathrm{Cl}/^{10}\mathrm{Be}$ ratio with increasing $\delta^{18}\mathrm{O}$ values (in ‰), as shown with the blue line in Figure 2C. The difference to a linear fit is small, but results in a slightly higher coefficient of determination ($R^2 = 0.52$ vs $R^2 = 0.47$) and a slightly lower residual sum of squares (RSS = 0.0035 vs RSS = 0.0038). Independent of the underlying processes, the relationship can be used to apply a first-order climate correction to the $^{36}\mathrm{Cl}/^{10}\mathrm{Be}$ ratio by dividing the decay corrected measured values by the $\delta^{18}\mathrm{O}$ predicted values. As shown in Figure 2D, this results in a lower standard deviation from the mean of $\sigma = 0.10$ than without the correction ($\sigma = 0.14$). This is an important improvement, since the standard deviation directly impacts the uncertainty of age estimates with the decay of the $^{36}\mathrm{Cl}/^{10}\mathrm{Be}$ ratio.

### 3.3 Age estimates

Age estimates for the deepest five samples were calculated with the exponential decay equation

$$\frac{^{36}\mathrm{Cl}}{^{10}\mathrm{Be}} = \left(\frac{^{36}\mathrm{Cl}}{^{10}\mathrm{Be}}\right)_0 e^{-kt}, \tag{2}$$

where $\frac{^{36}\mathrm{Cl}}{^{10}\mathrm{Be}}$ is the $\delta^{18}\mathrm{O}$ detrended ratio of undated samples, $\left(\frac{^{36}\mathrm{Cl}}{^{10}\mathrm{Be}}\right)_0$ is the $\delta^{18}\mathrm{O}$ detrended and decay corrected mean ratio of dated samples and $k$ is the decay constant given by $k = \frac{\ln(2)}{t_{1/2}}$ with an effective half-life of 384 kyr for the $^{36}\mathrm{Cl}/^{10}\mathrm{Be}$ ratio. The results are shown in Figure 3 together with the ST22 chronology (Hoffmann et al., 2022; Mulvaney et al., 2023). Note the break in the x-axis, which enables the display of the five previously undated samples with their respective top and bottom depths. Each sample spans over approximately 1.6 m depth and weighs about 1.3 kg (physical properties cut, circa 12 % of the ice core cross-section area (Grieman et al., 2024)). The orange shading indicates the age uncertainty related to the measurement uncertainty and the grey shading indicates the age uncertainties when additionally taking the uncertainty of the present-day value into account, inferred from the standard deviation of the decay corrected and $\delta^{18}\mathrm{O}$ detrended $^{36}\mathrm{Cl}/^{10}\mathrm{Be}$ ratio in Figure 2D. Age estimates with the latter uncertainty are quoted on the right hand side of the graph (see also Figure S1 in the supplementary information for a visualisation of the two uncertainty sources).

The $^{36}\mathrm{Cl}/^{10}\mathrm{Be}$ ratio of undated samples decreases monotonically with depth, suggesting a monotonically increasing age with depth. However, the $^{36}\mathrm{Cl}/^{10}\mathrm{Be}$ ratios of the first two undated samples at depths of 634.4 m and 636.8 m suggest ages of $47^{+77}_{-81}$ kyr BP and $105^{+81}_{-85}$ kyr BP, respectively, younger than the oldest ice of the published age scale, which ends with an age of 126 kyr BP at a depth of 627 m (Mulvaney et al., 2023). Considering the uncertainty, the deeper of the two samples may be older than 126 kyr, while the maximum age of the shallower sample would only be 124 kyr. We consider three possible explanations for the young ages:

1. A section of younger ice (at least 47 - 105 kyr BP) is repeated at the bottom of the core, similar to the repeated section of ice from 106 - 117 kyr BP (Mulvaney et al., 2023). This is unlikely, as the age range corresponds to about 70 m of ice




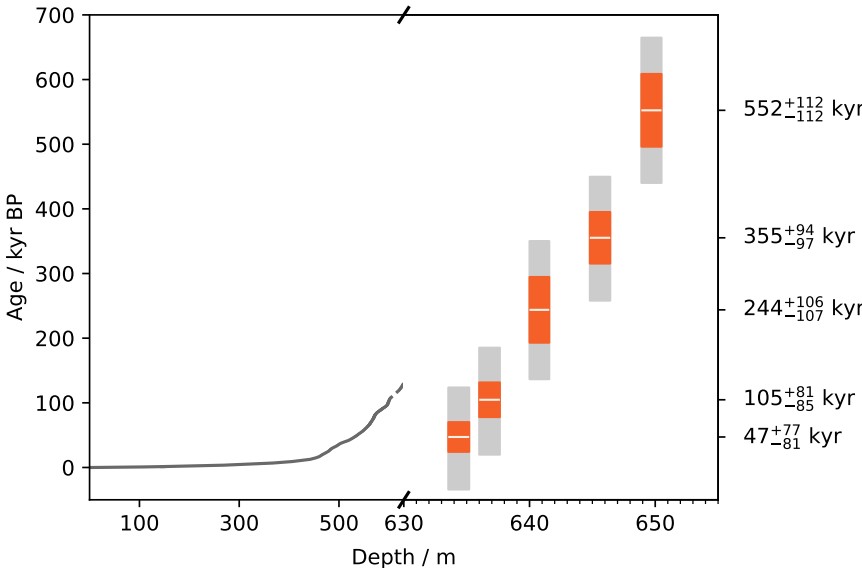

**Figure 3.** Age estimates for the five undated samples in the deepest section of the Skytrain ice core with combined analytical and present day value uncertainties.

from the last glacial period higher up in the core, while the repeated section in the Skytrain ice core and disturbances in Greenland ice cores affected only a few meters of ice associated with the last interglacial period (Mulvaney et al., 2023; Grootes et al., 1993).

2. Despite the comparably short length and young bottom age of the Skytrain ice core, $^{10}$Be may have migrated towards grain boundaries outside of the core, a process which is enabled through acidic liquid phases at grain boundaries and triple junctions (de Angelis et al., 2013; Fukazawa et al., 1998; Sakurai et al., 2017; Mulvaney et al., 1988). Similar behaviour has been postulated for the EDC, EDML, and GRIP ice cores (Kappelt et al., 2025; Raisbeck et al., 2006; Auer et al., 2009; Baumgartner et al., 1997a). Alternatively or additionally, recrystallisation may have resulted in new Be compounds which are not dissolved by our standard extraction method (Baccolo et al., 2021). If this is the case, all five ages would likely be underestimated due to apparent $^{10}$Be loss. Estimating the age with $^{36}$Cl alone yields older ages for four out of the five samples: $139^{+85}_{-97}$ kyr BP and $151^{+88}_{-100}$ kyr BP, respectively, for the shallowest two samples (see Figure S2 in the supplementary information). While changes in the production signal are not accounted for without including $^{10}$Be, the correction towards mostly younger ages hints towards a bias.

3. As shown in Figure 2D, the correction with $\delta^{18}$O does not completely remove climate-related variability from the $^{36}$Cl/$^{10}$Be ratio. Age estimate uncertainties are calculated based on one standard deviation, but some values deviate further from the mean. If the initial ratio of the shallowest sample was greater than the mean plus one standard deviation, the age would be underestimated.



It is very likely that all of the five samples are older than the last interglacial period and that the ages of the shallowest two samples are underestimated either due to apparent $^{10}$Be loss or due to unusually high initial $^{36}$Cl/$^{10}$Be ratios. With increasing age, the relative age uncertainty decreases, reaching about 20 % for the deepest sample with an age of $552^{+112}_{-112}$ kyr BP. This showcases that the method is best suited for samples with an age of at least one $^{36}$Cl half-life (308 kyr). The measurement and present-day value uncertainties contribute similarly to the overall uncertainty, so reductions of both would be equally helpful

for obtaining more precise age estimates.

    Previous work (Mulvaney et al., 2023) has shown that Skytrain Ice Rise had ice cover in the last interglacial, with ice 126 kyr old 24 m above the bed. With a date of at least 440 kyr (and most probably over 550 kyr) in ice spanning 0.5 - 2.1 m above the bed, we can state that the ice cap on Skytrain Ice Rise also survived marine isotope stage 11. Given the flow disturbance in ice from the last interglacial, it appears probable that, despite the monotonic increase in age with depth, there may be other

missing sections of ice, making further climatic interpretation of the old ice problematic.

## 4   Conclusions

A well-preserved peak of $^{36}$Cl from nuclear weapon tests in the 1950s and 60s was measured at the expected depth in the Skytrain ice core, confirming that $^{36}$Cl-loss is not an issue at the Skytrain ice rise and dating with the $^{36}$Cl/$^{10}$Be ratio is feasible. Age estimates are calculated from the degree of radioactive decay in a given sample in comparison to the present-day value,

which was approximated as the mean of 18 decay-corrected $^{36}$Cl/$^{10}$Be ratio measurements in ice of known age. A standard deviation of 14 % from the mean demonstrates a variability of the present-day value, which is a likely product of different transport and deposition pathways for $^{36}$Cl and $^{10}$Be. In part, the variability can be described with the $\delta^{18}$O signal, which negatively correlates with both, individual radionuclide concentrations and their ratio. Possible explanations are dilution and/or increased atmospheric depletion in warmer periods with higher precipitation rates and a higher scavenging efficiency for $^{36}$Cl

than for $^{10}$Be in mixed-phase clouds. We applied a correction to the $^{36}$Cl/$^{10}$Be ratio based on the $\delta^{18}$O signal to account for the climate influence and lowered the standard deviation of the present-day value to 10 %. Subsequently, we estimated the age of five discrete samples from ice below the dated section. While the relative uncertainty is large for samples younger than the half-life of $^{36}$Cl, it decreases for older samples, reaching 20 % at an age of 552 kyr. The measurement uncertainty and the uncertainty of the present day value contribute in about equal parts. Our results suggest that the ice cover at Skytrain Ice Rise

not only survived the last interglacial, but has been present since at least marine isotope stage 11.

*Data availability.* The radionuclide data are available at zenodo.org via https://doi.org/10.5281/zenodo.14209851 with license Creative Commons Attribution 4.0 International.



*Author contributions.* NK, EW, and RM designed the study; NK, EW, and RM acquired funding; NK, EW, MC, and CV performed the measurements; NK analysed and visualised the data; NK and EW wrote the article; all co-authors reviewed and edited the article.

*Competing interests.* One of the authors, Eric Wolff, is an editor of Climate of the Past.

*Acknowledgements.* This project has received funding from the European Union's Horizon 2020 research and innovation programme under the Marie Skłodowska-Curie grant agreement No 955750. The WACSWAIN project has received funding from the European Research Council under the Horizon 2020 research and innovation programme (grant agreement no. 742224, WACSWAIN). This material reflects only the authors' views and the Commission is not liable for any use that may be made of the information contained therein.



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
