# Peer review of "500-thousand-year-old basal ice at Skytrain Ice Rise, West Antarctica, estimated with the 36Cl/10Be ratio"

_EGUsphere, 2025_

## Referee Comment (RC1)

**Review of CP-2025-1780**

500-thousand-year-old basal ice at Skytrain Ice Rise, West Antarctic, estimated with the $^{36}$Cl/$^{10}$Be ratio", by N. Kappelt et al.

The authors present the results of a study to estimate the age of old ice at the bottom of the ~650 m deep Skytrain ice core in West Antarctica, using the 36Cl/10Be method. I think the introduction could use a bit of historical perspective, since this method was first pioneered by Nishiizumi et al. (1983) for Antarctic ice from Allan Hills and then used by Elmore et al. (1987) for Greenland ice. Later measurements by Nishiizumi and Finkel (1998; Chinese Science Bull 43) showed that the $^{36}$Cl/$^{10}$Be ratio varies systematically between ice cores (e.g. GISP2 vs. Siple Dome), so the method has not become a standard application to date old ice. The main challenge of the method is that it seems to require local calibration, as the authors have done in this study, so this seems a sound approach. Based on the measured 36Cl/10Be ratios in the top ~625 m of the core, which have been well dated, and a "climate correction" based on d18O, they used the climate-corrected $^{36}$Cl/$^{10}$Be ratios in the bottom 24 m of the core to estimate their ages, yielding values increasing with depth from ~50 kyr to 550 +/- 110 kyr. This old age at the bottom is an important finding for the climate record of the Skytrain ice core; and the observation that the age increases monotonically with depth gives some confidence in the method. The authors have done a good job in explaining the experimental uncertainties in the measured 36Cl/10Be ratios as well as the systematic uncertainty in the 36Cl/10Be ratio of young ice,

However, I would like to see a bit more discussion of the 36Cl/10Be ages of the dated samples. For example, one sample in Fig. 2c (not sure what depth) has a 36Cl/10Be ratio of ~0.19, what age would that correspond to ? In fact, it would be illustrative to plot the 36Cl/10Be-derived ages of all 18 samples in the top 620 m of the core in Figure 3, just to give a sense of how much the ages scatter – given the uncertainty of ~100 kyr in the ages of the deeper samples I would expect them to plot within ~200 kyr of their true age. This may also give some insight into how reliable the climate correction is.

Secondly, I would like to see a bit of discussion on the implications of this old ice at the bottom of the core. What does it mean to see a ~400 kyr increase in age over 24 m of ice thickness? I am not a glaciology expert, but I seem to remember from the WAIS Divide core that the projected age at the bottom of the core depended on the geothermal flux, i.e., more heating from the bottom means younger ice. So does the old ice imply a low geothermal flux and is this consistent with what we know about West Antarctica or is this beyond the scope of this paper ?

In summary, this study provides a valuable contribution for the ice core and climate change community and is an interesting result that will probably be tested by other methods. I recommend publication of this manuscript in Climate of the Past after minor revision. Besides the two comments above, I have a few small edits and suggestions (listed below) that may help to further improve the clarity of this paper.

Minor edits/suggestions/comments.
L22 – Explain where the effective half-life of 384 kyr comes from. Audi et al. (2017) lists a half-life of 301 kyr for $^{36}$Cl and 1.51 Myr for $^{10}$Be, whereas the updated value of Chmeleff et al. (2010) is 1.387 Myr, so it is not clear which one was used. Later in the paper (L198) a value of 308 kyr is quoted for $^{36}$Cl or is that a typo ?

L33-34. I'm sure the accumulation rate has varied in time, so may not always have been 13 g/cm$^2$/yr. So even though $^{36}$Cl has not been lost in the past 100 yr, is it possible that it may have been lost in the past when precipitation was lower ?

L78. Was there a particular reason to add more Cl carrier to the deeper samples ? Did the authors take the Cl component of the ice itself into account when converting 36Cl/Cl ratio to 36Cl concentration or is this contribution negligible compared to added carrier. If so, it would be useful to mention typical Cl concentration in Skytrain ice samples.

L198. Check 36Cl half-life – 301 kyr ?

---

## Author Comment (AC1)

We would like to thank Dr Welten for his review and his overall positive evaluation. We address his feedback, shown in blue, with our comments in black.

I think the introduction could use a bit of historical perspective, since this method was first pioneered by Nishiizumi et al. (1983) for Antarctic ice from Allan Hills and then used by Elmore et al. (1987) for Greenland ice. Later measurements by Nishiizumi and Finkel (1998; Chinese Science Bull 43) showed that the $^{36}Cl/^{10}Be$ ratio varies systematically between ice cores (e.g. GISP2 vs. Siple Dome), so the method has not become a standard application to date old ice.

Thank you for the suggestion, we agree that it will be a helpful addition to the introduction to inform the reader that this is not a standard dating method we apply, but that the paper should rather be seen as combination of method development and application. We will add: "The ratio was first suggested to be used as a dating tool by Nishiizumi et al. (1983) in Antarctica and by Elmore et al. (1987) in Greenland. However, due to its geospatial and temporal variability, it has not become a standard dating method. There are three processes that can cause its values in ice to deviate from the expected production rate ratio of 0.0875 (Poluianov 2016). [...] "

However, I would like to see a bit more discussion of the 36Cl/10Be ages of the dated samples. For example, one sample in Fig. 2c (not sure what depth) has a 36Cl/10Be ratio of ~0.19, what age would that correspond to? In fact, it would be illustrative to plot the 36Cl/10Be-derived ages of all 18 samples in the top 620 m of the core in Figure 3, just to give a sense of how much the ages scatter – given the uncertainty of ~100 kyr in the ages of the deeper samples I would expect them to plot within ~200 kyr of their true age. This may also give some insight into how reliable the climate correction is.

This is a good idea and gives a good overview of the scatter in age estimates, highlighting also that the method is only suited for older samples. Plotting all calculated ages in the original Figure shows that the estimated age is actually in agreement with the established chronology for all but two samples, the ~0.19 one you have mentioned in your comment, at a depth of about 550m, and one of the youngest samples around a depth of 100m. Additionally, looking at the age discrepancy between the radionuclide age and the chronology, all but this one sample are also within 100 kyr of the actual age. While transparently emphasising the uncertainty of the method, the addition of this data to the Figure also provides additional confidence in its validity in our opinion, so we are happy to include it. The slightly different age estimates for the five deep samples compared to the original manuscript are related to the inclusion of the ice's chlorine content, please see the answer to your comment about it further down.

[Figure]

Figure 1: Estimated ages for all samples and deviation from the ST22 chronology for samples below 630 m depth.

Secondly, I would like to see a bit of discussion on the implications of this old ice at the bottom of the core. What does it mean to see a ~400 kyr increase in age over 24 m of ice thickness? I am not a glaciology expert, but I seem to remember from the WAIS Divide core that the projected age at the bottom of the core depended on the geothermal flux, i.e., more heating from the bottom means younger ice. So does the old ice imply a low geothermal flux and is this consistent with what we know about West Antarctica or is this beyond the scope of this paper?

The reason there is not old ice at WAIS Divide is that there is melting at the bed, which removed the old ice. As stated, this was caused by a relatively high geothermal heat flux (GHF), and a thick ice sheet acting as a thermal insulator. However, at Skytrain the bed temperature is -15 degrees (Table 1 of Mulvaney et al 2021), so that there is no melting at the bed. That doesn't necessarily reflect a low GHF but rather is because the ice is thin. This means that the ice is frozen to the bed and thins to the bed (in the simplest case conforming to a Nye model where vertical strain rate is constant with depth and vertical velocity and annual layer thickness reach zero at the bed, but in reality to a more

complex solution). In such models the age increases rapidly towards the bed, as seen at Skytrain. We will reference the bed temperature in our discussion and explain that a rapid age increase towards the bed can be expected at Skytrain.

L22 – Explain where the effective half-life of 384 kyr comes from. Audi et al. (2017) lists a half-life of 301 kyr for $^{36}$Cl and 1.51 Myr for $^{10}$Be, whereas the updated value of Chmeleff et al. (2010) is 1.387 Myr, so it is not clear which one was used. Later in the paper (L198) a value of 308 kyr is quoted for $^{36}$Cl or is that a typo?

Good point, we will specify that the half life of 301 kyr for 36Cl listed by Audi and the updated 10Be half life of 1.387 Myr by Chmeleff are used. The value in L198 is a typo, it will be corrected to 301 kyr.

L33-34. I'm sure the accumulation rate has varied in time, so may not always have been 13 g/cm$^2$/yr. So even though $^{36}$Cl has not been lost in the past 100 yr, is it possible that it may have been lost in the past when precipitation was lower?

Indeed, the accumulation rate at many Antarctic drilling sites was about half of its present value in previous glacial times. Counter-intuitively, however, glacial conditions are more favourable towards chlorine preservation, as higher atmospheric concentrations of alkaline dust neutralised acidic species (HNO3, H2SO4, HCl). At low accumulation sites, like EPICA Dome C, both sea-salt chlorine and 36Cl were preserved, as we have explored in a previous publication (Quat Science Reviews, https://doi.org/10.1016/j.quascirev.2025.109254). We will include this information in the introduction.

L78. Was there a particular reason to add more Cl carrier to the deeper samples? Did the authors take the Cl component of the ice itself into account when converting 36Cl/Cl ratio to 36Cl concentration or is this contribution negligible compared to added carrier. If so, it would be useful to mention typical Cl concentration in Skytrain ice samples.

This is something we have overlooked. In our previous study with EDC ice, the contribution was negligible with natural chlorine concentrations of less than 1% of the added carrier mass. Here, however, we should include a correction, the average sample contains natural chloride with a weight of about 3.5% of the added carrier, due to the high sea-salt flux at this site. The correction modifies all data points and age estimates by a few percent, but does not affect our discussion of the data and the conclusions we draw. The correlations actually slightly improved and the standard deviation was further reduced from 0.10 to 0.09 for the detrended data.

[Figure]

Figure 2: Updated data considering natural chlorine.

The shallower samples used to analyse the bomb-peak were smaller than the deeper samples, so that the data could be obtained with a resolution of 1 - 2 years. While the deeper samples use standard amounts of carrier, less carrier was added to the shallower samples, which are smaller in size and, therefore, contain fewer radionuclide atoms. To maintain good counting statistics, less carrier was added with the trade-off of more difficult sample handling.

L198. Check 36Cl half-life – 301 kyr?

See above, this is a typo and will be corrected to 301 kyr.

---

## Author Comment (AC2)

We would like to thank the second referee for their review and address their comments in blue as follows:

While there is no doubt that it does not represent an issue at Skytrain ice rise during the Holocene or during interglacials, the authors should comment on what they expect to happen at this site during glacial periods, when the acc. rate is much lower.

It's a valid concern, the first reviewer made a similar suggestion, and it should definitely be addressed. Indeed, the accumulation rate at many Antarctic drilling sites was about half of its present value in previous glacial times. Counter-intuitively, however, glacial conditions are more favourable towards chlorine preservation, as higher atmospheric concentrations of alkaline dust neutralised acidic species ($HNO_3$, $H_2SO_4$, $HCl$). At low accumulation sites, like EPICA Dome C, both sea-salt chlorine and $^{36}Cl$ were preserved, as we have explored in a previous publication (Quat Science Reviews, https://doi.org/10.1016/j.quascirev.2025.109254). We will include this information in the introduction to emphasise that only the present-day accumulation rate has an influence on preservation and that chlorine is generally preserved under glacial conditions.

It's a good result that the 5 ages estimated in deep ice (figure 3) are getting older as depth increases but the discussion about the inconsistency of the first two points should be more detailed and the addition of the other experimental points (the younger ones) to this graph would greatly help in making clear how the estimated ages in the younger part relate to the official chronology.

We agree that it would be useful to include also the estimated ages for the younger samples and compare them with the established chronology to provide an overview of the scatter in age estimates and to highlight that the method is only suited for older samples. Plotting all calculated ages in the original Figure shows that the estimated age is actually in agreement with the established chronology for all but two samples. Additionally, looking at the age discrepancy between the radionuclide age and the chronology, all but this one sample are also within 100 kyr of the actual age. While transparently emphasising the uncertainty of the method, the addition of this data to the Figure also provides additional confidence in its validity in our opinion, so we are happy to include it. It also puts the comparably young age of the first two samples beyond the chronology into perspective, which, in reality, are probably about 130 kyr old.

[Figure]

Figure 1: Estimated ages for all samples and deviation from the ST22 chronology for samples below 630 m depth.

A short discussion about the influence of possible artifacts in the bottom ice seems to me very useful. Since 36Cl and 10Be have a different behaviour as concerning their movements in the ice, while the diffusion of H36Cl is properly discussed in the text, the possibility of an accumulation at grain boundaries of 10Be and 36Cl in the deep section should be briefly taken into account. If some information about the physical properties of the ice are available (crystal dimansions, etc), this should be mentioned in the paper to corroborate the meaning of the 36Cl/10Be ratio.

About possible migration of 10Be, we write in the original script: "Despite the comparably short length and young bottom age of the Skytrain ice core, 10Be may have migrated towards grain boundaries outside of the core, a process which is enabled through acidic liquid phases at grain boundaries and triple junctions (deAngelis et al. 2013, Fukazawa et al. 1998, Sakurai et al. 2017, Mulvaney et al. 1988}. Similar behaviour has been postulated for the EDC, EDML, and GRIP ice cores (Kappelt et al. 2025, Raisbeck et al. 2006, Auer et al. 2009, Baumgartner et al. 1997}. Alternatively or additionally, recrystallisation may have resulted in new Be compounds which are not

dissolved by our standard extraction method (Baccolo et al. 2021). If this is the case, all five ages would likely be underestimated."

We will add that no melting, which would favour migration, occurs at the bottom, as the bed temperature is -15 degrees (Table 1 of Mulvaney et al. 2021). Further research is needed to understand what happens to 10Be in deep ice, while there is no indication of migration or remineralisation for 36Cl. Even for 10Be there are only indicators, such as the older ages obtained with 36Cl alone and the shift from age overestimations to underestimations around a depth of 500 m with the ratio.

Line 77: are all the significant figures in 0.299 mg necessary?

We added Be and Cl carrier with the precision of three significant figures. While it is way within uncertainty, we prefer to report the carrier amount to the precision we aimed to achieve in this step.

Line 112: change to "... signal from Mulvaney et al. (2023)"

Thank you, we changed it.

Line 149: the numbers in the equation are slightly different from those in fig. 2c.

We will adapt the correct numbers of Fig 2(c) here.